# Exploring COVID-19 pandemic perceptions and vaccine uptake among community members and primary healthcare workers in Nigeria: A mixed methods study

Abiodun Sogbesan[1], Ayobami Adebayo Bakare[2,3]*, Sibylle Herzig van Wees[2], Julius Salako[1], Damola Bakare[1], Omotayo E. Olojede[1], Kofoworola Akinsola[1], Oluwabunmi Roseline Bakare[1], Adegoke G. Falade[1,4], Carina King[2]

**1** Department of Paediatrics, University College Hospital, Ibadan, Nigeria, **2** Department of Global Public Health, Karolinska Institutet, Stockholm, Sweden, **3** Department of Community Medicine, University College Hospital, Ibadan, Nigeria, **4** Department of Paediatrics, University of Ibadan, Ibadan, Nigeria

* bakare.ayobami.adebayo@ki.se

## Abstract

### Background

The COVID-19 pandemic significantly impacted global health, with diverse perceptions about the disease and its control measures, including vaccination. Understanding these perceptions can help inform public health and vaccination strategies in future outbreaks. This study examined community members and healthcare workers' (HCWs) perceptions of the COVID-19 pandemic and vaccines in Nigeria, exploring factors that influence vaccine acceptance and hesitancy.

### Methods

We conducted a mixed-methods study, combining quantitative survey data from 2,602 respondents (2,206 community members and 396 HCWs) with qualitative interviews (14 HCWs and 16 community members). Quantitative data were analyzed to identify factors associated with COVID-19 vaccine uptake and pandemic perceptions, while qualitative insights provided a deeper understanding of cultural perceptions, experiences, and hesitancy towards the COVID-19 vaccine.

### Results

Overall, 43.4% of community members and 96.7% of HCWs received at least one dose of the COVID-19 vaccine. Vaccine uptake was positively associated with increasing age, previous COVID-19 testing, male sex, government employment, and knowing someone diagnosed with COVID-19. Christianity was associated with lower uptake among community members. Perceptions varied, with 34.2% of community members and 17.7% of HCWs considering COVID-19 a death sentence, while 27.8%

**Data availability statement:** All relevant data are within the paper and its Supporting Information files.

**Funding:** While this study was supported by the Swedish Research Council (grant reference: 2020-04737), we confirm the funders had no roles in study design, data collection and analysis, decision to publish or preparation of the manuscript.

**Competing interests:** The authors have declared that no competing interests exist.

and 22.0% believed in 'African immunity,' respectively. Hesitancy was driven by the fear of side effects (32.6%), pregnancy-related concerns (25.9%), and convenience-related issues (13.5%). Qualitative data revealed misinformation, mistrust, fear of adverse reactions, logistical challenges, and belief in the sufficiency of childhood vaccination fueled hesitancy towards the COVID-19 vaccine. Despite this, general trust in vaccine safety and efficacy remained high, with most respondents willing to be vaccinated against other diseases and future outbreaks.

## Conclusion

Our findings underscore the need for tailored public health strategies to address specific sociodemographic factors, individual perceptions, and logistical challenges to enhance COVID-19 vaccine uptake. Public health campaigns should focus on debunking myths, improving vaccine literacy, and leveraging the social influence of respected community leaders to build trust.

## Introduction

Vaccines have contributed tremendously to improving global health, and their implementation is one of the most reliable and cost-effective interventions in public health that continues to save millions of lives every year [1–3]. However, vaccine hesitancy is one of the top 10 global health threats [4] and was prominent during the COVID-19 pandemic. After the discovery of the genome sequence of SARS-CoV-2 in early 2020, many COVID-19 vaccines were developed, with 50 of these vaccines receiving approval in 201 countries and 11 listed for emergency use by the WHO [5–9]. Subsequently, a global COVID-19 vaccination target was set for countries to vaccinate at least 70% of their populations by mid-2022 [9]. Vaccination of 100% of HCWs and 100% of the most vulnerable groups, with people aged over 60 years old and those who are immunocompromised or have underlying health conditions, is expected to be prioritized [9].

Despite the COVID-19 vaccine rollout and administration across all regions of the world, only 66.1% of the global eligible population had been vaccinated with a complete primary series as of 30 August 2023 [10]. While the Western Pacific (85.4%) and the Americas (71.2%) have surpassed the 70% vaccination target, the WHO African region has lagged behind, with only 32.4% of its population fully vaccinated [10]. Therefore, Africa has failed to achieve the African Union's goal of 70% population vaccine coverage by the end of 2022 [10,11]. In Nigeria, despite having seven government-approved vaccines, only 37.8% of eligible individuals have been fully vaccinated as of August 30, 2023 [10]. This represents a shortfall in meeting both international and the Federal Government vaccination coverage goals of 40% and 70% by the end of 2021 and 2022, respectively [12]. As of 2025, about 51.8% of the eligible population in Africa has been vaccinated, and Nigeria currently falls within the 40–69% coverage category for eligible populations [13]. This emphasizes ongoing efforts and the continued importance of understanding early barriers to vaccine uptake.

The reasons for low COVID-19 vaccination uptake are complex. The 3C model (convenience, complacency, and confidence) has been used to describe factors influencing COVID-19 vaccine uptake in sub-Saharan Africa [14,15]. Specifically, in Nigeria and across Africa, factors such as fears of side effects, ineffective public health communication, rumours and misinformation, and anxiety have contributed to low confidence in the safety and efficacy of COVID-19 vaccines [16–18]. Conversely, weaknesses in health systems, logistical gaps, inadequate funding, shortages of trained vaccinators, disruptions to essential health services, and concerns about vaccine accessibility impede the convenience and accessibility of COVID-19 vaccination efforts [18–20]. Finally, insufficient planning, apathy and disbelief in the existence of COVID-19 contributed to complacency or delays in COVID-19 vaccine administration [11,16,18].

In Nigeria, the factors influencing vaccine decision-making have not been sufficiently researched, particularly in the context of the COVID-19 pandemic. There is also limited evidence regarding the perceptions of COVID-19 vaccines among communities and HCWs. Investigating healthcare worker perceptions is essential, as they may significantly influence the effectiveness of vaccination promotion within the community. Therefore, we aimed to examine vaccination patterns, barriers, and drivers of COVID-19 vaccine uptake among Nigerians. This study allowed us to explore the complexities inherent in COVID-19 immunization programs and to learn lessons that will be relevant in the context of a new pandemic.

## Methods

### Study design

We conducted a mixed-methods parallel convergent study (QUAN+qual), comprising a facility-based cross-sectional quantitative survey and qualitative discussions with healthcare providers and community members. We included both healthcare workers and community members to gain a comprehensive understanding of COVID-19 vaccine uptake from the perspectives of both recipients and providers of immunization services. Quantitative analyses were conducted separately for the two groups to ensure clarity and preserve the distinctiveness of their experiences. For the qualitative component, which contributed a smaller portion of data to the mixed-methods integration, interviews were carried out with both groups, and coding and theme development were performed separately before being synthesized into overarching themes to explore system-wide insights. The study was carried out in three states in Nigeria: Oyo and Lagos in Southwestern Nigeria, and Jigawa in Northwestern Nigeria, from June to September 2022. This study forms part of a larger mixed-methods investigation, the "COVID-19 Vaccine Program Delivery in Nigeria," which examined the perspectives of healthcare providers and community members on COVID-19 vaccination program delivery, and how it influences and differs from the routine immunization program in Nigeria. Quantitative and qualitative results are presented together and triangulated under common headings.

### Study settings

The choice of Lagos, Oyo and Jigawa states was based on the COVID-19 burden and performance in routine and COVID-19 immunization programs. Lagos and Oyo ranked among the top five states with the highest COVID-19 burden [21], but were not among the top five performers during the early phases of the COVID-19 vaccine rollout [22]. Jigawa, despite not having a high COVID-19 burden, was the second top-performing state for the initial COVID-19 vaccine rollout [22].

Oyo and Lagos states are predominantly inhabited by the Yoruba ethnic group, with Christianity and Islam as the dominant religions. Jigawa state, located in the Northwest geopolitical zone, is predominantly inhabited by the Hausa and Fulani ethnic groups, with Islam being the most widely practiced religion. Although Lagos is the smallest of the three states in terms of land area, it has the largest population, estimated at 24.6 million [23]. Oyo and Jigawa, on the other hand, have estimated populations of 7.84 million and 6.83 million, respectively [24,25]. The study was conducted in Egbeda and Ibadan Southwest Local Government Areas (LGAs) in Oyo State, Kiyawa and Dutse LGAs in Jigawa State, and Ikeja and Ikorodu LGAs in Lagos State. These LGAs were purposively selected based on feasibility and accessibility.

## Study population

Data were collected from healthcare providers involved in immunization at the selected study sites. Community members included mothers who brought their children for routine immunization and adult recipients of COVID-19 vaccines in primary health facilities in the selected LGAs. Community members who had recently relocated, were not residents of LGA, or required urgent medical attention were excluded.

## Quantitative data collection and analysis

**Sample size determination.** The sample size for this study was determined based on the minimum sample size calculated in the wider study protocol for the COVID-19 Vaccine Program delivery in Nigeria, which estimated a single proportion: $n = Z\alpha^2 (p*(1-p)/d^2$. P represents the proportion of healthcare workers reporting post-vaccination side effects in Enugu, Southeast Nigeria (p = 87.6%) [26], with a precision of 5% (d = 0.05) and a confidence interval of 95% (z = 1.96). The estimated sample size was 186 participants per state after adjusting for a 10% non-response rate. Therefore, the total target sample size across the three states was 558 participants. However, the final total sample size was significantly larger than initially estimated (n = 2602), as more participants were recruited during data collection due to the use of convenience sampling and high attendance at participating health facilities.

**Sampling technique.** In each purposively selected LGA, we identified public and private facilities that offered COVID-19 vaccination services and routine immunization using the federal government health facilities database [27]. Facilities offering COVID-19 vaccinations were, where possible, matched by geography and ownership with facilities that provided only routine immunization and/or outpatient services during the data collection period. In Lagos and Oyo states, 11 PHCs offering COVID-19 vaccination and 11 PHCs offering routine immunization/outpatient services were selected. In Jigawa state, 11 primary healthcare facilities and 11 health posts offering COVID-19 vaccination were selected for each LGA. Overall, we collected data from 88 facilities across Lagos, Oyo and Jigawa State. All mothers and general adult outpatients who presented at the selected facilities were approached to participate using convenience sampling. In addition, all healthcare workers involved in immunization services at the selected facilities were purposefully selected.

**Data collection.** Data were collected using an interviewer-assisted questionnaire that was pre-tested in all three states. Trained data collectors with at least secondary education conducted in-person interviews to obtain information on respondents' sociodemographic characteristics, perceptions about COVID-19, COVID-19 experiences, including uptake of vaccination and side effects, perception about COVID-19 vaccines, and willingness to take other vaccines. Sociodemographic information assessed was age, sex, religion, ethnicity, marital status, household wealth index, monthly income, employment status, and education. Data was collected on Android tablets using Open Data Kit (ODK) software, and regular checks were performed to ensure accuracy.

**Study variables.** The primary outcome variable, "COVID-19 vaccine uptake," was defined as self-reported receipt of any dose of COVID-19 vaccine. Exposures of interest were categorized as follows: level of education (no formal education, primary, secondary, and tertiary), religion (Christianity and Islam), and government employment (yes/no). The household wealth index was analyzed using principal component analysis and categorized into tertiles.

**Data management and analysis.** We performed all quantitative data analyses using STATA version 16 (StataCorp LLC, College Station, TX, USA). We described respondents' characteristics, perceptions of COVID-19 disease, COVID-19 experiences (including vaccination patterns), reasons for COVID-19 vaccine acceptance and hesitancy, and perceptions about COVID-19 vaccines and willingness to take other vaccines, using frequencies, percentages, means and standard deviation. We used multivariable logistic regression to assess respondent factors associated with COVID-19 vaccine acceptance.

## Qualitative data collection and analysis

**Sample size and sampling.** Purposive and convenience sampling techniques were employed to recruit 14 HCWs who were involved in the national vaccination program (Jigawa: 8, and Oyo: 6) for in-depth semi-structured interviews. For

the community members, maximum variation sampling was employed to recruit 16 individuals (Jigawa: 8, and Oyo: 8), including those who had received zero, one, and two doses. This sample size was determined based on the expectation that the number would be sufficient to achieve saturation after conducting 9–17 interviews [28].

**Interview guide.**  The interview guides were developed based on our literature review [29,30]. The interview guide for healthcare providers included four sections: participants' socio-demographic information, their perception of COVID-19 vaccination, their understanding of COVID-19 vaccines and vaccination doses, and their experiences with COVID-19 vaccination (S1 Text). The interview guide for community members had three sections: participants' socio-demographic information, their perception of COVID-19 vaccination, and their vaccination experiences (S2 Text).

**Data collection.**  The research team comprised public health specialists. Interviews in Jigawa were conducted in English and Hausa by JS and two other female research nurses who had prior experience in qualitative data collection and were acquainted with the context. Interviews in Oyo were conducted in English and Yoruba by KOA and a research nurse with knowledge of the local setting. KOA is a female public health researcher with a Master of Public Health degree from Nigeria and experience in qualitative research. Interviews were conducted face-to-face in private locations for the participants and interviewers, which were home visits or workplaces for community members and primary healthcare facilities for healthcare workers. Each participant was given detergent as an incentive at the end of the interview. The interviews lasted between 45–60 minutes. Field notes were made during the interviews, and no repeat interviews were conducted.

**Data management and analysis.**  Interviews were audio-recorded, transcribed, and translated verbatim in English, and then stored on a secure cloud platform with restricted access to non-research team members. Thematic analysis was used to identify separate codes and themes for health providers and community members [31]. AAB and KOA independently coded the data. After the first round of coding, the codebook was compared and discussed. The coding team initially categorized and developed themes derived from the data, which were then shared with the entire research team for review.

**Reflexivity.**  The interviewers were non-indigenes who had no prior relationship with the participants but spoke the same language as the participants. AAB and KOA demonstrated cultural sensitivity and recognized that being based in Oyo State and understanding cultural nuances, societal norms, and beliefs could influence their interpretation of participants' responses. They also acknowledged the potential influence of their professional roles, training, experiences, and any preconceived notions about vaccination might affect their understanding of the participants' experiences. It was noted that AAB, a male community health physician, might bring a clinical perspective to the analysis by emphasizing individual health behaviours, patient-provider interaction, and the impact of medical misinformation on vaccine acceptance. KOA, a female public health researcher, approached the analysis from a broader public health perspective, focusing on systemic issues, such as healthcare and socio-economic determinants, and public health interventions to promote vaccine uptake. Continuous reflections on these different backgrounds were discussed within a small team and the larger research team.

### Ethical consideration

Ethical approval was obtained from the relevant ethics committees in all three states, including the UI/UCH Ethics Committee (ref: UI/EC/22/0139), the Oyo State Ministry of Health (ref: AD/13/479/44396A), the Lagos State Government (ref: LREC/06/10/1870), and the Jigawa State Ministry of Health (ref: JGHREC/2022/093). The study was conducted in compliance with the Declaration of Helsinki and the Nigerian National Code of Health Research Ethics. Verbal consent was obtained from respondents prior to their participation in the quantitative study, while written informed consent was obtained for the qualitative study. Participants were given the opportunity to review the informed consent form. In both studies, participants were informed that their involvement was voluntary and that the data collected would be used solely for research purposes.

## Results

### Study participants

We included 2602 participants in the quantitative analysis, 2206 (84.8%) community members and 396 (15.2%) health-care workers (S1 Fig.). Among community members, 83.9% were female, with a mean age of 33.2 years (SD ± 11.2). Over half of the community members (55.0%) belonged to the Yoruba ethnic group, 55.0% practiced Islam, 34.8% had secondary education, and 42.5% belonged to a 'poor' household. Among HCWs, 81.1% were female, with a mean age of 36.7 years (SD ± 10.3). Most of the HCWs (72.0%) belonged to the Yoruba ethnic group, 58.3% practiced Christianity, 83.3% had tertiary education, 50.5% belonged to 'wealthy' households, and 44.2% were not employed by the government despite working in public primary health facilities. (Table 1).

For the qualitative interviews, we recruited more females than males across community members (13 of 16 participants) and healthcare providers (8 of 14 participants). All six male healthcare providers were recruited from Jigawa State (S1 Table).

### Socio-demographic factors influencing vaccine acceptance

Analysis of vaccine uptake patterns showed a clear difference between community and HCW uptake, with 43.4% (957/2206) of community members receiving at least one dose of the COVID-19 vaccine, compared to 96.7% (383/396) of HCWs (Table 2). The proportion of individuals who had previously been tested for COVID-19 was considerably lower in both groups, with only 9.3% (206/2206) of community members and 33.0% (132/396) of HCWs having been tested. Additionally, 14.2% (314/2206) of community members and 29.0% (115/396) of HCWs knew someone previously diagnosed with COVID-19 (Table 2).

The odds of vaccination among community members and healthcare workers increased with increasing age [AOR = 1.04; 95% CI: 1.03–1.06] and [AOR = 1.06; 95% CI: 1.00–1.13], respectively, and with previous COVID-19 testing [AOR = 4.32; 95% CI: 3.13–5.98], and [AOR = 8.25; 95% CI: 1.04–65.28], respectively (Tables 3 and 4).

Among community members only, being male [AOR = 2.68; 95% CI: 1.54–4.66], having government employment [AOR = 2.46; 95% CI: 1.34–4.52], and knowing someone diagnosed with COVID-19 [AOR = 1.41; 95% CI: 1.06–1.87] were associated with higher odds of vaccination, while uptake was lower among community members who practiced Christianity [AOR = 0.73; 95% CI: 0.55–0.97] (Table 3).

### Individual perceptions influencing vaccine acceptance

Diverse perceptions regarding COVID-19 illness and vaccination were observed among community members and healthcare workers. Some community members questioned its existence, government involvement, and media portrayal, with 29.5% believing that COVID-19 was exaggerated by the media and 27.8% believing that Africans were immune to the virus. Additionally, 34.2% considered COVID-19 to be a death sentence. Healthcare workers also held differing views, with 22.0% believing in African immunity and 17.7% considering COVID-19 to be a death sentence (Table 5).

The primary reason for COVID-19 vaccine uptake cited by most community members (88.9%) and healthcare workers (96.1%) was to protect themselves against COVID-19. Compliance with government directives was reported by 36.5% of community members and 38.1% of healthcare workers. Additionally, 36.7% of healthcare workers reported taking the vaccine to encourage others, while a smaller proportion (1.3%) cited social influence (doing so because others were taking it). Some community members (15.3%) received the vaccine for travel purposes (Table 6).

While the quantitative data primarily highlighted self-protection and compliance with directives, qualitative insights delve deeper into individual factors. Personal health experiences and perceived vulnerability were central to decision-making. Across both states, participants emphasized both the immediate and long-term benefits of vaccine uptake. Participants in

**Table 1. Socio-demographic characteristics of respondents (N = 2602).**

| Characteristics | Community members/patients (N = 2206) | Healthcare workers (N = 396) |
|---|---|---|
| | N (%) | N (%) |
| **Age at recruitment (years)** | | |
| Mean (SD) | 33.2 (11.5) | 36.7 (10.3) |
| Minimum – maximum | 15–91 | 18–59 |
| **Sex** | | |
| Male | 354 (16.1) | 75 (18.9) |
| Female | 1852 (83.9) | 321 (81.1) |
| **Highest level of education** | | |
| No formal education | 474 (21.5) | 1 (0.2) |
| Primary | 239 (10.8) | 5 (1.3) |
| Secondary | 767 (34.8) | 58 (14.7) |
| Tertiary | 726 (32.9) | 330 (83.3) |
| Missing | – | 2 (0.5) |
| **Religion** | | |
| Christianity | 988 (44.8) | 231 (58.3) |
| Islam | 1213 (55.0) | 165 (41.7) |
| Other [a] | 4 (0.1) | – |
| Missing | 1 (0.1) | |
| **Ethnicity** | | |
| Yoruba | 1213 (55.0) | 285 (72.0) |
| Hausa | 597 (27.1) | 72 (18.2) |
| Igbo | 163 (7.4) | 11 (2.8) |
| Fulani | 111 (5.0) | 10 (2.5) |
| Other | 122 (5.5) | 18 (4.5) |
| **Employment by the government** | | |
| No | 2090 (94.7) | 175 (44.2) |
| Yes | 116 (5.3) | 221 (55.8) |
| **Monthly income (in naira)** | | |
| <30,000 (<minimum wage) | 677 (30.7) | 62 (15.7) |
| ≥30,000 | 318 (14.4) | 109 (27.5) |
| Declined | 1211 (54.9) | 225 (56.8) |
| **Wealth index (in tertiles)** | | |
| Lowest | 937 (42.5) | 79 (19.9) |
| Middle | 468 (21.2) | 110 (27.8) |
| Highest | 733 (33.2) | 200 (50.5) |
| Missing | 68 (3.1) | 7 (1.8) |

[a]Three respondents practice traditional religion, and one respondent with no religion.

Oyo State expressed reasons including setting examples for others, aligning with influential figures, ensuring job security, and protecting themselves and their families. For instance, one healthcare worker expressed,

> *"I saw our President; he took his own. Our state first man [the governor], he took his own. Our board, and my oga here took her own. So, who am I"?* (HCW 002 Oyo)

**Table 2. Respondents' COVID-19 experience (N = 2602).**

| Characteristics | Total | Community members (N=2206) | Healthcare workers (N=396) |
|---|---|---|---|
| | N (%) | N (%) | N (%) |
| **Had a COVID-19 test done before** | | | |
| Yes | 338 (13.0) | 206 (9.3) | 132 (33.3) |
| No | 2244 (86.2) | 1980 (89.8) | 264 (66.7) |
| Can't remember | 20 (0.80) | 20 (0.9) | – |
| **Had a positive COVID-19 result before (N=338)** | | | |
| Yes | 8 (2.4) | 2 (1.0) | 6 (4.5) |
| No | 321 (95.0) | 196 (95.1) | 125 (94.7) |
| Don't know | 9 (2.6) | 8 (3.9) | 1 (0.8) |
| **Know someone previously diagnosed with COVID-19 infection** | | | |
| Yes | 429 (16.5) | 314 (14.2) | 115 (29.0) |
| No | 2173 (83.5) | 1892 (85.8) | 281 (71.0) |
| **Received any dose of COVID-19 vaccines** | | | |
| Yes | 1340 (51.5) | 957 (43.4) | 383 (96.7) |
| No | 1247 (47.9) | 1234 (55.9) | 13 (3.3) |
| Don't know/unclassified | 15 (0.56) | 15 (0.7) | – |
| **Dose of COVID-19 vaccine received*** | | | |
| No dose | 1262 (48.5) | 1249 (56.6) | 13 (3.3) |
| First dose | 1340 (51.5) | 957 (43.4) | 383 (96.7) |
| Second dose | 896 (34.4) | 554 (25.1) | 342 (86.4) |
| Booster dose | 228 (8.8) | 97 (4.4) | 131 (33.1) |

Some community members noted,

*"Why I went was that government workers won't be able to work without it"* (Nursing mother 002 Oyo)

## Hesitancy and experiences with COVID-19 vaccination

The reasons for vaccine hesitancy were diverse and included confidence-related barriers, particularly fear of side effects (32.6%), pregnancy-related factors (25.9%), and convenience-related barriers (13.5%). Non-uptake of the second dose was driven by complacency-related barriers (51.6%), convenience-related barriers (30.9%), and pregnancy-related factors (16.8%). Non-uptake of booster doses was influenced by convenience-related barriers (32.5%), health system factors (29.8%), and information-related barriers (18.0%) (Fig 1 and S2 Table).

The qualitative analysis revealed varied experiences and expressions of COVID-19 vaccine hesitancy across study locations, shaped by contextual realities. In Jigawa, we did not find evidence of widespread hesitancy towards the COVID-19 vaccine; however, among those with COVID-19 vaccine-related sentiments, these were linked to misinformation, mistrust of the government, and social fears. Participants mentioned rumours and conspiracy theories about vaccine safety, including fears of infertility and population control:

*"Lack of trust in the government, they think they want to reduce the number of people in the world and Nigeria, some say it is for family planning"* (Caregiver 001 Jigawa)

Fear of adverse reactions or unverified incidents was cited to justify hesitancy within communities:

**Table 3. Factors associated with COVID-19 vaccine uptake among community members.**

| Characteristics | Crude OR (95% CI) | p-value | Adjusted OR [a] (95% CI) | p-value |
|---|---|---|---|---|
| **Age** | 1.05 (1.04–1.06) | **< 0.001** | 1.04 (1.03–1.06) | **< 0.001** |
| **Sex** | | | | |
| Female | 1 | | 1 | |
| Male | 3.96 (3.09–5.07) | **< 0.001** | 2.68 (1.54–4.66) | **< 0.001** |
| **Education** | | | | |
| Primary or less | 1 | | 1 | |
| Secondary | 1.25 (1.02–1.55) | **0.035** | 1.34 (0.69–2.60) | 0.387 |
| Tertiary | 1.95 (1.58–2.40) | **< 0.001** | 1.56 (0.82–2.96) | 0.172 |
| **Employed by government** | | | | |
| No | 1 | | 1 | |
| Yes | 5.14 (3.28–8.08) | **< 0.001** | 2.46 (1.34–4.52) | **0.004** |
| **Wealth index** | | | | |
| Lowest | 1 | | 1 | |
| Middle | 1.27 (1.02–1.60) | **0.036** | 0.88 (0.71–1.12) | 0.314 |
| Highest | 1.73 (1.42–2.11) | **< 0.001** | 1.01 (0.54–1.88) | 0.968 |
| **Religion** | | | | |
| Islam | 1 | | 1 | |
| Christianity | 1.13 (0.96. – 1.34) | 0.147 | 0.73 (0.55–0.97) | **0.032** |
| **Had a COVID-19 test done before** | | | | |
| No | 1 | | 1 | |
| Yes | 4.95 (3.54–6.92) | **< 0.001** | 4.32 (3.13–5.98) | **< 0.001** |
| **Know someone previously diagnosed with COVID-19** | | | | |
| No | 1 | | 1 | |
| Yes | 2.25 (1.76–2.87) | **< 0.001** | 1.41 (1.06 −1.87) | **0.020** |

[a]Adjusted for state clustering.

*"It has reaction; some will be sleepy all day and night, some fever, some dizziness"* (Caregiver 003 Jigawa)

*"There was a community we went to for vaccination, and they said, a man died after being vaccinated, and the community ever since then refused to receive vaccine [COVID-19 vaccine]"* (HCW 04 Jigawa)

Besides these sentiments, some community members required motivation through material or financial incentives to receive the vaccine. Cultural dynamics also shaped hesitancy, with some women indicating that permission from their husbands was required before seeking vaccinations.

*"Some of us will have to take permission from our husbands"* (Nursing mother 001 Jigawa)

In contrast, participants in Oyo generally expressed a greater willingness to accept the COVID-19 vaccine; however, logistical and operational barriers hindered its uptake. Long waiting times and concerns about job disruption were recurring challenges.

*"I took an excuse from my workplace that I want to go and take covid vaccine, right? On getting there, I was told I'll wait for 3/4 hours. Do you think I will wait? I won't like to risk my job"* (Community member 002 Oyo)

**Table 4. Factors associated with COVID-19 vaccine uptake among healthcare workers.**

| Characteristics | Crude OR (95% CI) | p-value | Adjusted OR (95% CI) | p-value |
|---|---|---|---|---|
| **Age** | 1.04 (0.98–1.11) | 0.153 | 1.06 (1.00–1.13) | **0.046** |
| **Sex** | | | | |
| Female | 1 | | 1 | |
| Male | 2.87 (0.37–22.4) | 0.314 | 0.93 (0.10–8.94) | 0.948 |
| **Religion** | | | | |
| Islam | 1 | | 1 | |
| Christianity | 0.41 (0.11–1.51) | 0.180 | 0.59 (0.15–2.33) | 0.453 |
| **Had a COVID-19 test done before** | | | | |
| No | 1 | | 1 | |
| Yes | 6.23 (0.80–48.50) | 0.080 | 8.25 (1.04–65.28) | **0.046** |
| **Know someone previously diagnosed with COVID-19** | | | | |
| No | 1 | | 1 | |
| Yes | 0.64 (0.21–2.01) | 0.450 | 0.40 (0.11–1.43) | 0.159 |
| **State** | | | | |
| Lagos | 1 | | 1 | |
| Oyo | 0.45 (0.14–1.52) | 0.200 | 0.32 (0.09–1.14) | 0.079 |
| Jigawa[a] | – | – | – | – |

[a]There was a perfect prediction as all were vaccinated.

While misinformation did exist in Oyo, it appeared less widespread and was more often linked to mistrust of political leadership than to social fears or health myths.

*"There is wrong information on COVID-19 vaccine, wrong news of side effects and so on. Some people believed that government is using the COVID-19 vaccine to steal more money for their families"* (HCW 03 Oyo)

Some participants also questioned the need for additional vaccines, citing childhood immunisations as already sufficient.

*"I don't like it; I believe that all the vaccines that I have taken in my childhood are enough to sustain me"* (Nursing mother 03 OYO)

### Trust in vaccines and future intentions regarding vaccination

Despite COVID-19 vaccine hesitancy, particularly among community members, a significant proportion of respondents expressed overall trust in vaccine safety and efficacy, indicating a willingness to vaccinate against other diseases. There was little difference between unvaccinated and vaccinated respondents in their reported belief in the safety (94.9% vs. 98.2%) and efficacy (95.5% vs. 98.2%) of routine vaccines, reflecting their perceptions before the COVID-19 pandemic. Among those who received the COVID-19 vaccine, belief in vaccine safety and efficacy increased to 99.2% and 99.0%, respectively. Furthermore, the majority of vaccine recipients (99.3%) expressed willingness to vaccinate their children against childhood illnesses, 97.8% were willing to take a vaccine against diseases like hepatitis, 97.0% were willing to take a vaccine against human papillomavirus (HPV), and 98.7% were willing to take vaccines in the future in case of another similar outbreak (S3 Table).

**Table 5. Respondents' perception about COVID-19 (N = 2602).**

| Characteristics | Total | Community members (N = 2206) | Healthcare workers (N = 396) |
|---|---|---|---|
| **COVID-19 is not real** | | | |
| Agree | 526 (20.2) | 467 (21.2) | 59 (14.9) |
| Disagree | 1828 (70.3) | 1496 (67.8) | 332 (83.8) |
| Don't know | 248 (9.5) | 243 (11.0) | 5 (1.3) |
| **There is no COVID-19 in Nigeria** | | | |
| Agree | 367 (14.1) | 336 (15.2) | 31 (7.8) |
| Disagree | 1925 (74.0) | 1568 (71.1) | 357 (90.2) |
| Don't know | 310 (11.9) | 302 (13.7) | 8 (2.0) |
| **COVID-19 is a scam by the government** | | | |
| Agree | 334 (12.8) | 312 (14.1) | 22 (5.6) |
| Disagree | 1698 (65.3) | 1361 (61.7) | 337 (85.1) |
| Don't know | 570 (21.9) | 533 (24.2) | 37 (9.3) |
| **COVID-19 is exaggerated by the media** | | | |
| Agree | 714 (27.4) | 651 (29.5) | 63 (15.9) |
| Disagree | 1335 (51.3) | 1044 (47.3) | 291 (73.5) |
| Don't know | 553 (21.3) | 511 (23.2) | 42 (10.6) |
| **COVID-19 is God's punishment to mankind** | | | |
| Agree | 806 (31.0) | 730 (33.1) | 76 (19.2) |
| Disagree | 1194 (45.9) | 941 (42.7) | 253 (63.9) |
| Don't know | 602 (23.1) | 535 (24.2) | 67 (16.9) |
| **COVID-19 is a biological weapon** | | | |
| Agree | 470 (18.1) | 418 (18.9) | 52 (13.1) |
| Disagree | 1343 (51.6) | 1063 (48.2) | 280 (70.7) |
| Don't know | 789 (30.3) | 725 (32.9) | 64 (16.2) |
| **The virus was designed by the pharmaceutical industry to sell drugs** | | | |
| Agree | 293 (11.3) | 272 (12.3) | 21 (5.3) |
| Disagree | 1523 (58.5) | 1207 (54.7) | 316 (79.8) |
| Don't know | 786 (30.2) | 727 (33.0) | 59 (14.9) |
| **Africans are immune to COVID-19** | | | |
| Agree | 699 (26.9) | 612 (27.8) | 87 (22.0) |
| Disagree | 1462 (56.2) | 1174 (53.2) | 288 (72.7) |
| Don't know | 441 (16.9) | 420 (19.0) | 21 (5.3) |
| **COVID-19 is a disease of the elites** | | | |
| Agree | 488 (18.8) | 448 (20.3) | 40 (10.1) |
| Disagree | 1785 (68.6) | 1437 (65.1) | 348 (87.9) |
| Don't know | 329 (12.6) | 321 (14.6) | 8 (2.0) |
| **COVID-19 can affect both young and old** | | | |
| Agree | 2150 (82.6) | 1793 (81.3) | 357 (90.1) |
| Disagree | 242 (9.3) | 210 (9.5) | 32 (8.1) |
| Don't know | 210 (8.1) | 203 (9.2) | 7 (1.8) |
| **COVID-19 is a death sentence** | | | |
| Agree | 825 (31.7) | 755 (34.2) | 70 (17.7) |
| Disagree | 1412 (54.3) | 1102 (50.0) | 310 (78.3) |
| Don't know | 365 (14.0) | 349 (15.8) | 16 (4.0) |

**Table 6. Reasons for COVID-19 vaccine acceptance among respondents who received any dose of vaccination.**

| Reasons for taking vaccine[a] | Community members/ patients (n = 957) | Healthcare workers (n = 383) |
|---|---|---|
| To protect me against COVID-19 infection | 832 (86.9) | 368 (96.1) |
| To comply with the government directive | 349 (36.5) | 146 (38.1) |
| To encourage others to take it | 117 (12.2) | 144 (37.6) |
| For travel purposes | 146 (15.3) | 66 (17.2) |
| Because others are taking it | 33 (3.5) | 5 (1.3) |
| Because it is free | 31 (3.2) | 11 (2.9) |
| Have had COVID-19 infection before | 1 (0.1) | 3 (0.8) |
| To comply with the workplace/boss's directive | 10 (1.0) | 1 (0.3) |
| Because I will be paid/given cash | 2 (0.2) | – |
| Others | 6 (0.6) | 2 (0.5) |

[a]Respondents could provide multiple responses.

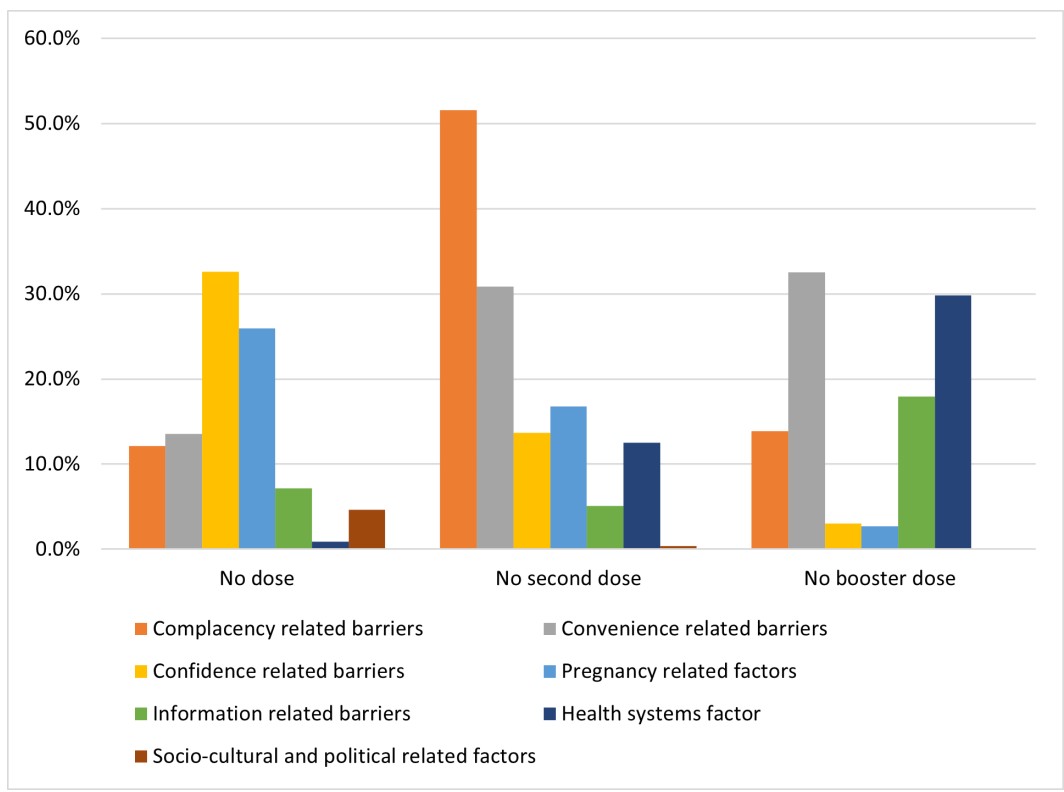

**Fig 1. Reasons for COVID-19 vaccine hesitancy among unvaccinated eligible respondents.**

## Discussion

In this study, we examined COVID-19 vaccination patterns and factors influencing vaccine uptake and perceptions of the COVID-19 pandemic among community members and healthcare workers in Nigeria. We found significant disparities in vaccination rates and highlighted the multifaceted nature of vaccine acceptance and hesitancy. While HCWs exhibited high initial vaccine uptake, the drop-off was high, with only one-third completing the booster dose. In contrast, over half of the community members remain unvaccinated, highlighting persistent challenges in achieving comprehensive COVID-19 vaccination coverage, particularly in the WHO African region, which has not attained the 70% vaccination target [10]. These findings are consistent with previous reports of suboptimal COVID-19 vaccine uptake in Nigeria [32–34].

We found several socio-demographic factors positively associated with vaccine acceptance among community members, including older age, male sex, a history of COVID-19 testing, and knowing someone previously diagnosed with COVID-19. These findings align with the understanding that older individuals at an increased risk of severe COVID-19 outcomes are more likely to seek vaccination [35]. Knowing someone affected by COVID-19 served as a motivational factor for vaccination, likely due to heightened awareness of the disease's severity [36]. For HCWs, older age and previous COVID-19 testing were associated with vaccine uptake, suggesting that direct exposure to COVID-19, whether through professional experience or personal testing, significantly influences vaccine acceptance among HCWs [36]. The heightened vulnerability of older HCWs likely drives their proactive health behaviours, including vaccination.

Interestingly, our adjusted model showed lower COVID-19 vaccine uptake among Christian community members compared to their Muslim counterparts. This finding may reflect context-specific factors such as differences in vaccine messaging, denominational beliefs, and levels of trust in government or health systems. The existing literature presents a complex picture. A regional survey from Africa and the Asia-Pacific aligns with our findings, indicating that African Muslims were more likely than Christians to support COVID-19 vaccinations [37]. In contrast, global-level quantitative analyses have linked lower vaccine coverage with both Muslim populations and some Christian denominations, such as Eastern Orthodox, while Catholics and non-religious groups have demonstrated higher acceptance [38]. Furthermore, a qualitative study among Pentecostal churches in Tanzania revealed mixed attitudes, with some groups showing hesitancy and others accepting vaccination. It highlighted that individual decisions were often influenced more by social and environmental factors than by religious doctrine alone [39].

We also found that male community members had significantly higher odds of receiving the COVID-19 vaccine compared to female community members. This likely reflects a mix of structural, social, and cultural factors. In many parts of Nigeria, men tend to have greater autonomy in health-related decision-making and better access to health information, which may promote timely vaccination. By comparison, women, as observed in our study and in UNICEF case studies from Nigeria, often face barriers such as the need for spousal consent, limited mobility due to childcare responsibilities, and restrictive cultural norms [40]. Furthermore, lower levels of female education may reduce awareness of vaccination benefits [40]. These factors likely contributed to the lower uptake observed among women, despite their high representation in our study sample.

Individual perceptions also play a significant role in shaping vaccine acceptance. Both community members and HCWs expressed scepticism about the existence of COVID-19, notions of African immunity, and misconceptions that the media exaggerated the disease. Prior studies have shown that public perceptions and beliefs significantly influence compliance with preventive measures and contribute to vaccine hesitancy [41–43]. A study conducted by the Africa CDC on COVID-19 vaccine perception across 15 countries, including Nigeria, found misconceptions, mistrust, misinformation, and fears about vaccine safety [44]. Rumours and myths surrounding vaccine development by developed nations as a population reduction strategy in Africa, including claims that these vaccines cause infertility, preceded COVID-19 and have fostered mistrust among the public [33,45–48]. These findings underscore the need for comprehensive strategies to correct misbeliefs and misinformation and highlight the importance of targeted engagement with community heads and religious leaders, who are highly regarded in African communities, to address misconceptions and enhance vaccine confidence.

COVID-19 vaccination programs were a cornerstone of the pandemic response, serving both to curb outbreaks and as long-term instruments to prevent subsequent waves of the COVID-19 pandemic [49]. Key obstacles to vaccination identified in our quantitative findings included confidence-related issues, particularly fear of side effects, as documented in previous studies [33,46,47], as well as pregnancy-related concerns and convenience-related barriers. The qualitative data revealed additional barriers, such as inconveniences related to vaccine administration, the need for spousal approval, and belief in the sufficiency of childhood vaccinations. Confidence barriers tended to diminish with subsequent vaccine doses, suggesting a gradual increase in trust over time. However, respondents reported higher levels of complacency, convenience-related issues, health system factors, and information barriers for subsequent doses than for the initial dose. Complacency was particularly high for the second dose, indicating a need for continuous awareness campaigns. Convenience-related barriers, such as prolonged waiting times and limited access, were significant deterrents, consistent with the findings of Agha (2021), who reported that only 32% of respondents found it easy to obtain a COVID-19 vaccination for themselves [50], reinforcing the need for streamlined vaccination processes.

Despite these barriers, we identified several motivating factors, including protection against infection, compliance with government directives, encouragement from others, and travel purposes. We also observed the influence of social factors, such as seeing influential figures like the president and local leaders receive the vaccine, which motivated others to follow suit. Among HCWs, receiving the vaccine to set an example for others and fulfil job requirements was particularly evident. The findings of this study align with those of a qualitative study conducted in Ethiopia, where religious leaders who are highly respected in their communities, not only promoted preventive measures during the pandemic but also received vaccinations publicly to build trust among the people [48].

## Limitations

We had four key limitations in our study. First, the reliance on self-reported data may have introduced bias, as participants might have over-reported and under-reported their vaccination status, perceptions, and experiences due to social desirability or recall bias. However, COVID-19 vaccination in Nigeria was not associated with significant stigma, so social desirability bias may have been minimal. Second, data collection was limited to community members and healthcare workers at primary healthcare facilities, meaning the perspectives and experiences of other community members without facility engagement during the data collection period were not explored. Third, while our qualitative sample was designed to achieve saturation and capture diverse perspectives across different vaccine uptake levels and professional roles, the relatively small number of participants may not fully reflect the wide spectrum of experiences across Nigeria's highly heterogeneous population. Finally, logistical constraints and variations in vaccine availability during the study period could have influenced vaccination patterns, thus affecting the results. Despite these limitations, our mixed-method approach provided a unique opportunity to understand the intricacies of COVID-19 immunization programs in Nigeria and glean insights applicable to future pandemics.

## Conclusion

Our study highlights preventable obstacles affecting vaccination intentions and emphasizes the need for the Nigerian Government to strengthen its implementation program. This includes improving vaccine literacy through effective public communication strategies and ensuring consistent vaccine accessibility. The findings underscore the importance of addressing religious beliefs and socio-demographic factors that significantly influence vaccine uptake. Furthermore, overcoming barriers such as misinformation, mistrust of the government, and logistical challenges is essential. Engaging influential figures and trusted individuals in shaping messages and information sharing could help build confidence and ensure the spread of accurate information. Future research should also explore behavioural and psychological drivers of vaccine decisions, including media influence and trust in health institutions, to broaden the scope of analysis and inform more targeted interventions. These lessons are critical for preparing for future pandemics.

## Supporting information

**S1 Fig. Participants inclusion in data collection at facilities.**
(PDF)

**S1 Table. Qualitative – Sociodemographic characteristics of respondents.**
(PDF)

**S2 Table. Reasons for COVID-19 vaccine hesitancy among unvaccinated eligible respondents.** [a] Respondents could provide multiple responses, [b] Excluding respondents that are not eligible for the dose, dose not required or did not indicate reasons.
(PDF)

**S3 Table. Respondents' perception about COVID-19 vaccine, recommendation & willingness to take other vaccines.**
(PDF)

**S1 Text. Healthcare provider In-depth Interview guide.**
(PDF)

**S2 Text. Community members In-depth Interview guide.**
(PDF)

**S1 Dataset. Raw dataset.**
(XLSX)

**S2 Dataset. Raw data set dictionary.**
(CSV)

**S3 Dataset. Data set for Figure 1.**
(XLSX)

## Acknowledgments

We thank the data collectors, healthcare workers and community members for their time and support.

## Author contributions

**Conceptualization:** Abiodun Sogbesan, Ayobami Adebayo Bakare, Sibylle Herzig Van Wees, Carina King.

**Data curation:** Abiodun Sogbesan, Ayobami Adebayo Bakare, Julius Salako, Damola Bakare, Omotayo E Olojede.

**Formal analysis:** Abiodun Sogbesan, Ayobami Adebayo Bakare, Sibylle Herzig Van Wees, Kofoworola Akinsola, Carina King.

**Funding acquisition:** Adegoke G Falade, Carina King.

**Investigation:** Abiodun Sogbesan, Ayobami Adebayo Bakare, Julius Salako, Damola Bakare, Omotayo E Olojede, Kofoworola Akinsola, Oluwabunmi Roseline Bakare, Adegoke G Falade.

**Methodology:** Abiodun Sogbesan, Ayobami Adebayo Bakare, Sibylle Herzig Van Wees, Damola Bakare, Kofoworola Akinsola, Adegoke G Falade, Carina King.

**Project administration:** Abiodun Sogbesan, Ayobami Adebayo Bakare, Omotayo E Olojede, Adegoke G Falade.

**Resources:** Abiodun Sogbesan, Oluwabunmi Roseline Bakare.

**Supervision:** Julius Salako, Omotayo E Olojede.

**Validation:** Sibylle Herzig Van Wees.

**Visualization:** Carina King.

**Writing – original draft:** Abiodun Sogbesan, Ayobami Adebayo Bakare, Julius Salako, Kofoworola Akinsola.

**Writing – review & editing:** Sibylle Herzig Van Wees, Damola Bakare, Omotayo E Olojede, Oluwabunmi Roseline Bakare, Adegoke G Falade, Carina King.

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
