## [Decision Letter · Decision Letter 0]

23 Feb 2025

Dear Dr. Bakare,

Thank you for submitting your manuscript to PLOS ONE. After careful consideration, we feel that it has merit but does not fully meet PLOS ONE’s publication criteria as it currently stands. Therefore, we invite you to submit a revised version of the manuscript that addresses the points raised during the review process.

We look forward to receiving your revised manuscript.

Kind regards,

Morufu Olalekan Raimi, Ph.D

Academic Editor

PLOS ONE

**Journal Requirements:**

1. When submitting your revision, we need you to address these additional requirements. Please ensure that your manuscript meets PLOS ONE's style requirements, including those for file naming. The PLOS ONE style templates can be found at https://journals.plos.org/plosone/s/file?id=wjVg/PLOSOne_formatting_sample_main_body.pdf and https://journals.plos.org/plosone/s/file?id=ba62/PLOSOne_formatting_sample_title_authors_affiliations.pdf 2. We note that the grant information you provided in the ‘Funding Information’ and ‘Financial Disclosure’ sections do not match.  When you resubmit, please ensure that you provide the correct grant numbers for the awards you received for your study in the ‘Funding Information’ section. 3. Thank you for stating the following financial disclosure: Link Grant (Swedish Research Council).  Please state what role the funders took in the study.  If the funders had no role, please state: "The funders had no role in study design, data collection and analysis, decision to publish, or preparation of the manuscript." If this statement is not correct you must amend it as needed. Please include this amended Role of Funder statement in your cover letter; we will change the online submission form on your behalf. 4. We note that you have indicated that there are restrictions to data sharing for this study. For studies involving human research participant data or other sensitive data, we encourage authors to share de-identified or anonymized data. However, when data cannot be publicly shared for ethical reasons, we allow authors to make their data sets available upon request. For information on unacceptable data access restrictions, please see http://journals.plos.org/plosone/s/data-availability#loc-unacceptable-data-access-restrictions.  Before we proceed with your manuscript, please address the following prompts: a) If there are ethical or legal restrictions on sharing a de-identified data set, please explain them in detail (e.g., data contain potentially identifying or sensitive patient information, data are owned by a third-party organization, etc.) and who has imposed them (e.g., a Research Ethics Committee or Institutional Review Board, etc.). Please also provide contact information for a data access committee, ethics committee, or other institutional body to which data requests may be sent. b) If there are no restrictions, please upload the minimal anonymized data set necessary to replicate your study findings to a stable, public repository and provide us with the relevant URLs, DOIs, or accession numbers. Please see http://www.bmj.com/content/340/bmj.c181.long for guidelines on how to de-identify and prepare clinical data for publication. For a list of recommended repositories, please see https://journals.plos.org/plosone/s/recommended-repositories. You also have the option of uploading the data as Supporting Information files, but we would recommend depositing data directly to a data repository if possible. Please update your Data Availability statement in the submission form accordingly. 5. PLOS requires an ORCID iD for the corresponding author in Editorial Manager on papers submitted after December 6th, 2016. Please ensure that you have an ORCID iD and that it is validated in Editorial Manager. To do this, go to ‘Update my Information’ (in the upper left-hand corner of the main menu), and click on the Fetch/Validate link next to the ORCID field. This will take you to the ORCID site and allow you to create a new iD or authenticate a pre-existing iD in Editorial Manager. 6. Please include captions for your Supporting Information files at the end of your manuscript, and update any in-text citations to match accordingly. Please see our Supporting Information guidelines for more information: http://journals.plos.org/plosone/s/supporting-information. 7. We note that this data set consists of interview transcripts. Can you please confirm that all participants gave consent for interview transcript to be published? If they DID provide consent for these transcripts to be published, please also confirm that the transcripts do not contain any potentially identifying information (or let us know if the participants consented to having their personal details published and made publicly available). We consider the following details to be identifying information:- Names, nicknames, and initials- Age more specific than round numbers- GPS coordinates, physical addresses, IP addresses, email addresses- Information in small sample sizes (e.g. 40 students from X class in X year at X university)- Specific dates (e.g. visit dates, interview dates)- ID numbers Or, if the participants DID NOT provide consent for these transcripts to be published:- Provide a de-identified version of the data or excerpts of interview responses- Provide information regarding how these transcripts can be accessed by researchers who meet the criteria for access to confidential data, including:a) the grounds for restrictionb) the name of the ethics committee, Institutional Review Board, or third-party organization that is imposing sharing restrictions on the datac) a non-author, institutional point of contact that is able to field data access queries, in the interest of maintaining long-term data accessibility.d) Any relevant data set names, URLs, DOIs, etc. that an independent researcher would need in order to request your minimal data set. For further information on sharing data that contains sensitive participant information, please see: https://journals.plos.org/plosone/s/data-availability#loc-human-research-participant-data-and-other-sensitive-data If there are ethical, legal, or third-party restrictions upon your dataset, you must provide all of the following details (https://journals.plos.org/plosone/s/data-availability#loc-acceptable-data-access-restrictions):a) A complete description of the datasetb) The nature of the restrictions upon the data (ethical, legal, or owned by a third party) and the reasoning behind themc) The full name of the body imposing the restrictions upon your dataset (ethics committee, institution, data access committee, etc)d) If the data are owned by a third party, confirmation of whether the authors received any special privileges in accessing the data that other researchers would not havee) Direct, non-author contact information (preferably email) for the body imposing the restrictions upon the data, to which data access requests can be sent?

**Additional Editor Comments:**

The reviewers have provided detailed and constructive feedback on your manuscript, "Exploring COVID-19 Pandemic Perceptions and Vaccine Uptake among Community Members and Primary Healthcare Workers in Nigeria: A Mixed Methods Study." Their comments highlight several areas for improvement, including the need to clarify methodological details (e.g., sample size calculations, sampling techniques), expand the discussion of key findings (e.g., gender, religion, and regional influences on vaccine uptake), and address limitations related to sample diversity and representation. Additionally, incorporating recent studies and exploring additional variables (e.g., transportation challenges, media influence) will strengthen the manuscript's relevance and depth. Please ensure that citation formatting is corrected throughout the document. Addressing these points will enhance the manuscript's contribution to the field and provide a more nuanced understanding of vaccine perceptions and hesitancy in Nigeria. I look forward to receiving your revised submission.

Reviewers' comments:

Reviewer's Responses to Questions

**Comments to the Author**

1. Is the manuscript technically sound, and do the data support the conclusions?

Reviewer #1: Yes

Reviewer #2: Partly

Reviewer #3: Yes

2. Has the statistical analysis been performed appropriately and rigorously?

Reviewer #1: Yes

Reviewer #2: No

Reviewer #3: No

3. Have the authors made all data underlying the findings in their manuscript fully available?

Reviewer #1: Yes

Reviewer #2: Yes

Reviewer #3: Yes

4. Is the manuscript presented in an intelligible fashion and written in standard English?

Reviewer #1: Yes

Reviewer #2: Yes

Reviewer #3: No

**Reviewer #1:**  Introduction: Provides a good background to the study in line with what the situation was at the time the data was collected. However, now there are many similar studies that have since been published and it would be desirable to include some of the recent studies.

Methods and Results:

Appropriately described and implemented as stated.

The sample size stated on page 5-6 was estimated at 558 of the three states, and on page 10 the sample size reported is 2602. Need clarification in the way this statement is worded, was 558 for each of the 3 State or what?

A very high number of female respondents (83.9%) – why? And what impact did this have on the overall results/study outcome?

Even in qualitative respondents, majority were women, why?

Page 16, vaccine uptake was lower among community members who practiced Christianity. Why?

Discussion:

1). Expound on your findings in the discussion section by helping readers understand why some of the results came out as they did:

• Examples:

o Why being male was associated with better/higher.

o Being of Christianity faith was associated with low uptake.

2). The finding of faith leaders in Ethiopia associated with low uptake (page 26 second paragraph) contradicts with last sentence in the discussion (page 27 – 28) from the same reference/citation. The authors need to make this clear for the readers. Which one is which? What is true from the reference/citation?

3). It is helpful to also include studies that are contradictory to some of your findings or those that are inclusive and provide a synthesis of how your study fits into the big picture when you integrate such. It helps strengthen the study but also provides a springboard for further research discussions and research. Gives an all round picture of the issue in the context of study and beyond.

Conclusion: Done well, supported with findings/data from the study.

Overall Comments

1). The paper gives a balanced report based on its mixed methods approach.

2). ***Citations are made after the “period” this is wrong. As stated above. This appears throughout the document. Should be corrected.

3). With the specific suggestions above, the paper provides good information similar to many other studies on this topic as it has been one of the most highly researched issue with the emergence of COVID-19 globally.

**Reviewer #2:**  I appreciate the submission and this is an important topic to addressed and subsequenlty being published. However, one of the major concerns arrive from this submisision is the combination of the community and HWC. The community and HWCs are of two various background and experienced differences with COVID-19. There was no justification was why both of these groups were combined into the analysis. If the intention is to do comparison, the data analysis failed to capture this. The strenghten of this manuscript is in it mixed-methods study, which strengthen the study. Given that the number of community is large, perhaps the authors could focus on just this community and leave the HWCs data for a separate submission.

**Reviewer #3:** The study explored COVID-19 pandemic perceptions and vaccine uptake among community members and primary healthcare workers (HCWs) in Nigeria. 43.4% of community members and 96.7% of HCWs received at least one dose of the COVID-19 vaccine. 27.8% of community members and 22.0% of HCWs believed in "African immunity" to COVID-19. The author should address the following points……

1. The study primarily focused on community members and healthcare workers (HCWs) at primary healthcare facilities, potentially overlooking insights from individuals who were not engaged with these facilities. Relying on self-reported vaccine uptake and perceptions may have led to inaccurate results due to memory gaps and the tendency to give socially acceptable answers.

2. While household wealth index and government employment were analyzed, factors like transportation challenges, internet access (for misinformation) and local economic impact of COVID-19 should explored.

3. Only 14 healthcare providers and 16 community members were interviewed, which might not fully capture the diverse range of opinions and experiences in a country as large as Nigeria.

4. The study was conducted in only three Nigerian states (Oyo, Lagos, and Jigawa), which may not fully represent national trends in vaccine hesitancy and uptake.

5. The qualitative analysis did not extensively compare the perceptions across different regions, ethnic groups, or social classes in Nigeria.

6. The study integrates qualitative and quantitative findings, but there are gaps in both analyses. A mixed-methods approach was useful, but the study could have strengthened the correlation by expanding the qualitative sample size for greater diversity.

7. Although spousal approval and pregnancy-related concerns were identified, a deeper gender-based analysis was lacking to explore how social roles and expectations influenced vaccine decision-making

8. The study did not extensively explore behavioral or psychological factors influencing vaccine uptake beyond demographics and basic perceptions. More detailed variables, such as media influence, trust in healthcare institutions, or personal experiences with previous vaccinations, were not deeply analyzed. Including these parameters will expand the scope of variables in enhancing quantitative analysis.

9. In sampling technique indicate whether the study used random sampling, convenience sampling or stratified sampling.

**Do you want your identity to be public for this peer review?** For information about this choice, including consent withdrawal, please see our Privacy Policy

Reviewer #1: No

Reviewer #2: No

Reviewer #3: **Yes:** Ekramul Islam

---

## [Author Response · Author response to Decision Letter 1]

5 Jul 2025

We thank the reviewers for their thoughtful comments. We have now addressed them and provided responses to all the comments from the three reviewers in a separate document uploaded as "Response to reviewers"

---

## [Editor Report · Decision Letter 1]

14 Jul 2025

Dear Dr. Bakare,

Thank you for submitting your manuscript to PLOS ONE. After careful consideration, we feel that it has merit but does not fully meet PLOS ONE’s publication criteria as it currently stands. Therefore, we invite you to submit a revised version of the manuscript that addresses the points raised during the review process.

We look forward to receiving your revised manuscript.

Kind regards,

Morufu Olalekan Raimi, Ph.D

Academic Editor

PLOS ONE

**Journal Requirements:**

**Additional Editor Comments:**

Final Editorial Decision: Accept with Minor Revisions

Updated Editorial Comments:

Thank you for submitting the revised manuscript and your point-by-point responses to reviewer feedback. Upon evaluation of the revised submission and full manuscript, the following observations and conclusions inform this editorial decision:

Strengths of the Revised Manuscript:

1. Methodological Clarity Improved:

o The sampling strategy is now well-articulated, including the rationale for combining community members and HCWs in the overall study design, while analyses were disaggregated appropriately in the results.

o Sample size explanation has been clarified (558 as the minimum estimate vs. 2602 actually recruited).

o The study follows COREQ-consistent reporting, including reflexivity, ethical safeguards, data collection transparency, and saturation justification.

2. Integration of Quantitative and Qualitative Findings:

o The study now integrates qualitative insights more effectively with quantitative data (e.g., reasons for hesitancy, social factors).

o Findings are triangulated across respondent groups and states (Jigawa vs. Oyo), providing contextual depth.

3. Reviewer Concerns Addressed:

o Reviewer 1's query about the high proportion of female respondents was directly addressed as a function of the sampling context (mothers at immunization clinics).

o Reviewer 2’s concern about combining HCWs and community members is mitigated by separate quantitative analyses and justifications in methods and discussion.

o Reviewer 3’s comments on limitations, regional variation, and lack of deeper behavioral variables were acknowledged and discussed explicitly in the updated Limitations section.

4. Technical and Ethical Compliance:

o Ethical approvals, consent procedures, and data availability align with PLOS ONE standards.

o Formatting and citation structure appear corrected (e.g., references moved inside punctuation).

Minor Issues to Correct Before Final Acceptance:

1. Grammar and Clarity:

o A light copyedit pass is still advised to correct minor typographical and grammatical inconsistencies (e.g., “COVID-19 is God’ns punishment”).

o Ensure uniform tense and parallel sentence structures, especially in thematic summaries.

2. Figures and Appendices:

o Figure 1 (on vaccine hesitancy) should be rechecked for clarity and consistency with textual descriptions. Ensure legends and categories match exactly what is described in the body.

o Make sure Supplementary Tables and Interview Guides (S1–S3) referenced in the manuscript are uploaded and appropriately linked.

3. Reference Formatting:

o Double-check reference consistency (e.g., spacing, author initials) to ensure alignment with PLOS ONE citation style across all 50+ entries.

Conclusion and Next Steps:

The manuscript demonstrates methodological rigor, policy relevance, and clear public health utility in Nigeria’s COVID-19 context. The authors have been responsive to reviewer critiques and have improved clarity, validity, and contextual depth throughout the revision.

Therefore, I recommend accepting this manuscript after minor editorial corrections.

Action Required:

• Submit a final clean version with the noted corrections.

• Ensure all supporting files (S1–S3 Tables, Interview Guides, COREQ Checklist) are uploaded correctly.

• A final editorial check will confirm readiness for publication.

We appreciate the authors’ diligent work and thoughtful revisions. Congratulations on this important contribution.

Sincerely,

Manuscript ID: PONE-D-24-36637

---

## [Author Response · Author response to Decision Letter 2]

4 Sep 2025

Responses to the Academic Editor

1. Grammar and Clarity:

We sincerely thank the editor for these helpful comments. We have carefully revised the manuscript to address both points and ensured that the requested changes have been implemented consistently across all relevant sections.

o A light copyedit pass is still advised to correct minor typographical and grammatical inconsistencies (e.g., “COVID-19 is God’ns punishment”).

Thank you for this observation. We have thoroughly copyedited the manuscript to correct all typographical and grammatical inconsistencies, including the example noted. A detailed review was carried out to ensure clarity and accuracy throughout the text.

o Ensure uniform tense and parallel sentence structures, especially in thematic summaries.

We appreciate this important feedback. We have carefully revised the manuscript to ensure uniform tense and parallel sentence structures across all sections, particularly in the thematic summaries. These changes have improved the overall readability and academic precision of the manuscript.

2. Figures and Appendices:

o Figure 1 (on vaccine hesitancy) should be rechecked for clarity and consistency with textual descriptions. Ensure legends and categories match exactly what is described in the body.

We thank the reviewer for this observation. Upon review, we found that the previously attached legends and categories included elements from findings that were neither represented in Figure 1 nor described in the textual narrative. To ensure clarity and complete alignment with the text, we have removed these extraneous legends and categories on page 23, paragraph 2. The revised Figure 1 caption now contains only the title, ensuring consistency between the figure and the textual descriptions.

o Make sure Supplementary Tables and Interview Guides (S1–S3) referenced in the manuscript are uploaded and appropriately linked.

Thank you for this reminder. The supplementary files (S1–S3 Tables and Interview guides: S1-S2 Texts) have already been uploaded. However, as we submit these current responses, we will confirm once again that the files are correctly uploaded and appropriately linked to ensure there are no issues.

3. Reference Formatting:

o Double-check reference consistency (e.g., spacing, author initials) to ensure alignment with PLOS ONE citation style across all 50+ entries.

Thank you for this observation. We have carefully reviewed and revised all references to ensure full compliance with the PLOS ONE citation style. Specific corrections were made to references 4, 9, 10, 11, 12, 13, 24, and 49 to address formatting inconsistencies related to author initials, spacing, and punctuation.

Additionally, reference 23 has been entirely replaced. The previous reference, which provided a projected population estimate for Lagos State, was outdated and has since been shown to be inaccurate by other sources. We have now replaced it with a more recent reference from the United Nations Environment Programme (UNEP), which reflects the current and more reliable population estimate for Lagos State.

---

## [Editor Report · Decision Letter 2]

22 Sep 2025

Dear Dr. Bakare,

Thank you for submitting your manuscript to PLOS ONE. After careful consideration, we feel that it has merit but does not fully meet PLOS ONE’s publication criteria as it currently stands. Therefore, we invite you to submit a revised version of the manuscript that addresses the points raised during the review process.

We look forward to receiving your revised manuscript.

Kind regards,

Morufu Olalekan Raimi, Ph.D

Academic Editor

PLOS ONE

Journal Requirements:

Additional Editor Comments:

“Exploring COVID-19 Pandemic Perceptions and Vaccine Uptake among Community Members and Primary Healthcare Workers in Nigeria: A Mixed Methods Study.”

The study is strong, relevant, and well-structured. However, there are numerous grammatical, stylistic, and formatting issues that need to be addressed to meet the high publication standards of a journal like PLOS ONE. The authors have responded to the editor's points, but the manuscript itself still requires significant copyediting.

Here is a detailed, section-by-section review and critique.

Overall Impression

• Strengths: The research question is highly relevant. The mixed-methods approach is appropriate and well-executed. The sample size is large and diverse. The findings are significant for public health policy in Nigeria and similar contexts.

• Weaknesses: The manuscript is marred by persistent grammatical errors, inconsistent tense usage, awkward phrasing, and minor formatting issues. These errors detract from the clarity and professional presentation of the otherwise excellent science.

Line-by-Line Critique

Title & Authorship (Page 1 & 11):

• Generally fine.

Abstract (Page 12):

• Line 2 (Page 12): healthcare workers' (HCWs) -> healthcare workers (HCWs)' (The apostrophe should come after the acronym).

• Line 8 (Page 12): fuelled -> fueled (PLOS ONE uses American English spelling).

• The abstract is well-written and summarizes the key findings effectively.

Introduction (Pages 13-15):

• Page 14, Line 1: 66.1% of the eligible population had been vaccinated -> 66.1% of the global eligible population had been vaccinated (Add “global” for clarity).

• Page 14, Line 10: This emphasizes -> This emphasizes (American English).

• Page 15, Line 1: complexities inherent in COVID-19 immunization programmes; and to learn -> complexities inherent in COVID-19 immunization programs and to learn (Remove semicolon; American English: “programs”).

Methods (Pages 15-21):

• Study Design (Page 15):

o Preservationaintain -> preserve and maintain (Typo).

o programmes; -> programs (American English, remove semicolon).

o it exerts influences on-and differs from -> it influences and differs from (Awkward phrasing).

• Study Settings (Page 16):

o land arcamass -> land area (Typo).

o largesthighest population, with an estimated at population of18-20million–24.6 million -> largest population, estimated at 24.6 million (Major typo/editing error. The correct figure from the new UNEP source is 24.6M? The text is garbled and must be fixed).

o based on their feasibility -> based on feasibility (Remove “their”).

• Study Population (Page 16):

o healthcare providers involved in immunization afin the selected study sites -> healthcare providers involved in immunization at the selected study sites (Typo).

o facilities inof the selected LGAs -> facilities in the selected LGAs (Typo).

o were not residents of LGA residents -> were not residents of the LGA (Repetitive).

• Quantitative Data Collection (Page 17-18):

o Data waswere collected -> Data were collected (“Data” is plural).

o Sociodemographic information assessed were was -> Sociodemographic information assessed was (Subject-verb agreement).

o performed forto ensure -> performed to ensure (Typo).

o Household Household wealth index -> The household wealth index (Duplicate word).

o including the pattern of vaccination, and reasons -> including vaccination patterns, reasons (Awkward insertion).

• Qualitative Data Collection (Page 19-20):

o we interviewed 16 individuals (Jigawa: 8, and Oyo: 8), with maximum variation sampling to include those who received zero, 1, and 2 doses. This sentence is repetitive with the one before it. Combine or delete.

o included four sections: focused on -> included four sections: (Remove "focused on”).

o who have had prior experience -> who had prior experience (Tense).

o aware acquainted -> were acquainted (Typo).

o KOA wasis -> KOA is (Tense).

o stored-in on a secure cloud -> stored on a secure cloud (Typo).

o double-blinded to each other’s coding -> "Double-blinded" is unusual in qualitative research. Simpler: coded the data independently.

o which was were then shared withexpanded to -> which were then shared with (Grammar/typo).

• Reflexivity (Page 20-21):

o spoke the same language as them–participants -> spoke the same language as the participants (Awkward).

o sensitivity; and recognizing -> sensitivity and recognized (Punctuation/tense).

o shape influence -> influence (Choose one).

o thate potential influence of their professional roles -> the potential influence of their professional roles (Typo).

o affecton their understanding of their–participants’ experiences -> affect their understanding of the participants’ experiences (Typos/awkward phrasing).

o AAB, as a male community health physician, might bring a clinical perspective to the analysis; by emphasizing -> AAB, a male community health physician, might bring a clinical perspective to the analysis, emphasizing (Punctuation).

o including such as healthcare -> such as healthcare ("including such as" is redundant).

o as well as and public health interventions into promoting -> and public health interventions to promote (Grammar/typo).

o thesedifferingdifferent backgrounds -> these different backgrounds (Redundant).

Results (Pages 21-35):

• Page 21, Participant Description: The majority of the Among community members -> Among community members, the majority (Editing error).

• Page 22, Table 1: *Three respondents practise traditional religion -> *Three respondents practice traditional religion (American English).

• Page 24, Table 2: having been tested (undergone testing. -> having been tested. (Editing error).

• Page 25, Text: increased with increasing age [AOR = 1.04;...] and [AOR = 1.06;...], respectively, as well as withand previous COVID-19 testing -> increased with increasing age... and with previous COVID-19 testing (Typo/awkward phrasing).

• Page 25, Text: Uptake was lower among those community members who practised Christianity -> Uptake was lower among community members who practiced Christianity (Typo).

• Page 28, Table 5: COVID-19 is God’ns punishment -> COVID-19 is God's punishment (Grammatical error noted by the editor, still present).

• Page 31, Table 6: respondents that-who received -> respondents who received (Typo).

• Page 32, Qualitative Quotes: “I saw our President; he took his own... So, who am I” -> “I saw our President; he took his own... So, who am I?” (Add closing question mark).

• Page 33, Fig 1 Caption: The note in parentheses is critical context that should likely be moved into the main text or the figure legend itself to ensure clarity, as per the editor's request.

• Page 33, Text: these were linked concerns were related to -> these concerns were related to (Repetitive).

• Page 33, Text: Participants mentioned expressed-rumours -> Participants expressed rumours OR Participants mentioned rumours (Typo).

• Page 33-34, Text: receive the vaccine;- -> receive the vaccine. (Replace semicolon with period).

o required before seeking vaccinations;- -> required before seeking vaccinations.

o were recurring challenges;- -> were recurring challenges.

• Page 34, Text: vaccine is are enough -> vaccines are enough (Subject-verb agreement).

• Page 34, Text: unvaccinated and vaccinated respondents–in their -> unvaccinated and vaccinated respondents in their (Remove stray hyphen).

• Page 34, Text: increased with COVID-19-to 99.2% -> increased to 99.2% (Editing error).

Discussion (Pages 35-39):

• Page 35, Line 1: underscored highlighted -> highlighted (Redundant).

• Page 35, Line 3: highlightingunderscoring persistentongoing challenges -> highlighting persistent challenges (Redundant).

• Page 35, Line 5: has failed to meet not attained -> has not attained (Redundant).

• Page 36, Line 3: appeared to served as -> served as (Grammar).

• Page 36, Line 8: vaccination-than Christians -> vaccination than Christians (Typo).

• Page 37, Line 1: have demonstratedshowed higher acceptance -> showed higher acceptance (Redundant).

• Page 37, Line 2: AdditionallyFurthermore -> Furthermore (Choose one).

• Page 37, Line 3: emphasizedhighlighted -> highlighted (Redundant).

• Page 37, Line 4: influenced more influenced by -> influenced more by (Repetitive).

• Page 37, Line 9: mayean facilitatepromote -> may promote (Editing error).

• Page 37, Line 10: bothas observed in -> as observed in (Typo).

• Page 37, Line 11: the need forrequiring -> requiring (Redundant).

• Page 37, Line 12: and-limited mobility -> limited mobility (Typo).

• Page 37, Line 13: FurthermoreAdditionally -> Additionally (Choose one).

• Page 37, Line 13: mayean reduceaffect -> may reduce (Editing error).

• Page 37, Line 18: thatabout the media exaggerated the disease-being exaggerated -> that the media exaggerated the disease (Awkward phrasing).

• Page 38, Line 1: identifiedfound -> found (Redundant).

• Page 38, Line 2: includingwith claims -> including claims (Grammar).

• Page 38, Line 3: have fosteredled-to mistrust -> have fostered mistrust (Awkward).

• Page 38, Line 4: correctrectify -> rectify (Redundant).

• Page 38, Line 6: and among their congregations -> This is a new idea. Consider integrating it better: ...religious leaders, who are highly regarded in African communities, to address misconceptions among their congregations and enhance...

• Page 38, Line 9: identifiedreported -> identified (Redundant).

• Page 38, Line 10: particularlyespecially -> particularly (Redundant).

• Page 38, Line 12: revealedprovided -> revealed (Redundant).

• Page 38, Line 18: indicating athe need -> indicating a need (Typo).

• Page 39, Line 1: identifiedfound -> identified (Redundant).

• Page 39, Line 3: observedsw the influence -> observed the influence (Typo).

• Page 39, Line 4: receivetake the vaccine -> receive the vaccine (Choose one).

• Page 39, Line 5: receivingtaking the vaccine -> receiving the vaccine (Choose one).

• Page 39, Line 7: and congregations not only -> not only (Remove “and congregations”; it breaks the sentence flow. The point is that leaders promoted measures).

• Page 39, Line 12: curutilizing a mixed-method approach in our study provided -> utilizing a mixed-methods approach provided OR our mixed-methods approach provided (Typo/awkward phrasing).

• Page 39, Line 13: understandcomprehend -> understand (Redundant).

• Page 39, Line 14: futureduring another pandemics -> future pandemics (Editing error).

References (Pages 40-45, 87-93):

• The authors state they have fixed references 4, 9, 10, 11, 12, 13, 24, 49 and replaced 23. This must be verified in the final submitted manuscript file. The reference list in the provided text appears correct, but the final submission must be checked meticulously for PLOS ONE style (e.g., journal abbreviations, use of “et al.”, punctuation, etc.).

Response to Reviewers (Page 94-95):

• The responses are professional and address the editor's points adequately.

Summary of Major Recommendations

1. Thorough Copyedit: The manuscript requires a complete, word-by-word copyedit by a native English speaker or professional editing service focused on:

o Grammar and syntax.

o Tense consistency (often shifts between past and present).

o American English spelling (e.g., emphasize, practice (verb), program).

o Removing redundant phrases and awkward constructions.

2. Consistency Check: Ensure all instances of “programme” are changed to “program”. Ensure all demographic data (especially the Lagos population figure) is accurate and consistently reported.

3. Figure 1 Legend: Integrate the explanatory note from the caption (about “other reasons” and “conspiracy theories”) into the main results text for clarity, as this directly addresses the editor's point about aligning the figure with the text.

4. Final Proofread: After implementing all changes, the manuscript must be proofread one final time to catch any remaining errors before submission.

This study makes a valuable contribution. By meticulously addressing these language and presentation issues, the authors will significantly strengthen their manuscript and improve its chances of acceptance.

Sincerely,

Dr. Morufu Olalekan RAIMI,

BSc, (Geography. and Environmental Management), Diploma. (Environmental Health), M.Sc. Environmental Health Management), M.Phil. (Environmental Health Science), P.hD (Environmental Health Science), MNES, REHO, LEHO, FAIWMES

Environmental Health Consultant/Lecturer at Federal University Otuoke, Bayelsa State. Nigeria.

Environmental Health Consultant to United Nations Economic Commission for Europe (UNECE) Expert Group on Resources Management (EGRM). Geneva, Switzerland.

Research Consultant to Bayelsa State Primary Health Care Board.

Former Technical Adviser to the Executive Secretary, Bayelsa State Primary Health Care Board.

Former Director, Advocacy, Communication and Social Mobilization, Bayelsa State Primary Health Care Board.

Program Manager, Centre for Niger Delta Studies and Sustainability (CNDSS), Federal University Otuoke, Bayelsa State.

Deputy Director, Niger Delta Institute for Emerging and Re-emerging Infectious Diseases (NDIERID), Federal University Otuoke, Bayelsa State.

Reviewer to National Science Foundation (NSF) Graduate Research Fellowship Program (GRFP)

https://publons.com/a/1479339/

https://ssrn.com/author=2891311

ORCID iD: https://orcid.org/0000-0001-5042-6729

Web of Science Researcher ID: https://publons.com/a/1479339/

Website: https://ssrn.com/author=2891311;
https://www.growkudos.com/profile/morufu_raimi;
https://sciprofiles.com/profile/Morufuolalekanraimi;
https://livedna.org/234.27529

https://scholar.google.com/citations?user=nRBW82AAAAAJ&hl=en.

https://theconversation.com/profiles/morufu-olalekan-raimi-1520774

---

## [Author Response · Author response to Decision Letter 3]

12 Nov 2025

We thank the reviewers and editors for their useful feedback to improve the readability of this article. We have made required suggestions as necessary based on the feedback received. However, we would like to highlight that most of the typographical errors identified could have been misinterpreted if the editors had reviewed the track changes version. Nevertheless, we had reviewed and confirmed that all queries raised have been addressed.

Thank you very much for your time

---

## [Editor Report · Decision Letter 3]

18 Nov 2025

Exploring COVID-19 Pandemic Perceptions and Vaccine Uptake among Community Members and Primary Healthcare Workers in Nigeria: A Mixed Methods Study.

PONE-D-24-36637R3

Dear Authors,

We’re pleased to inform you that your manuscript has been judged scientifically suitable for publication and will be formally accepted for publication once it meets all outstanding technical requirements.

Kind regards,

Morufu Olalekan Raimi, Ph.D

Academic Editor

PLOS ONE

Additional Editor Comments (optional):

Manuscript ID: PONE-D-24-36637R3

Title: Exploring COVID-19 Pandemic Perceptions and Vaccine Uptake among Community Members and Primary Healthcare Workers in Nigeria: A Mixed Methods Study

Dear Dr. Bakare,

Thank you for submitting the revised version of your manuscript to PLOS ONE. We appreciate the time and effort you have invested in addressing the reviewers' comments and improving the quality of your work.

After careful re-evaluation of the revised manuscript and your detailed responses to the reviewers, I am pleased to inform you that the manuscript has now met the publication criteria for PLOS ONE.

Summary of Review:

The reviewers initially raised several important points regarding methodological clarity, sample representation, discussion of findings (particularly concerning gender, religion, and regional influences), and the integration of qualitative and quantitative data. You have satisfactorily addressed these concerns in the revised submission. Key improvements include:

• Clarification of sample size calculations and sampling techniques.

• Enhanced discussion of socio-demographic and cultural factors influencing vaccine uptake.

• Appropriate handling of mixed-methods integration and reflexivity.

• Corrections to citation formatting and language consistency as per PLOS ONE style.

Your revisions have strengthened the manuscript's rigor, clarity, and contribution to the literature on COVID-19 vaccine perceptions in Nigeria.

Additional Notes:

• All data availability and ethical requirements have been adequately addressed.

• The financial disclosure and funding information are now consistent.

• The manuscript is now written in clear, standard English.

We congratulate you on your work and its contribution to the field.

Next Steps:

Your manuscript has been accepted for publication in PLOS ONE. You will receive further instructions regarding production and proofing shortly.

Thank you for choosing PLOS ONE, and we look forward to publishing your research.

Sincerely,

Morufu Olalekan Raimi, Ph.D
---

## [Editor Report · Acceptance letter]

PONE-D-24-36637R3

PLOS One

Dear Dr. Bakare,

I'm pleased to inform you that your manuscript has been deemed suitable for publication in PLOS One. Congratulations! Your manuscript is now being handed over to our production team.

Kind regards,

on behalf of

Prof Morufu Olalekan Raimi

Academic Editor

PLOS One